# TPO: Tree Search Policy Optimization for Continuous Action Spaces

## Abstract

Monte Carlo Tree Search (MCTS) has achieved impressive results on a range of discrete environments, such as Go, Mario and Arcade games, but it has not yet fulfilled its true potential in continuous domains. In this work, we introduce TPO, a tree search based policy optimization method for continuous environments. TPO takes a hybrid approach to policy optimization. Building the MCTS tree in a continuous action space and updating the policy gradient using off-policy MCTS trajectories are non-trivial. To overcome these challenges, we propose limiting tree search branching factor by drawing only a few action samples from the policy distribution and defining a new loss function based on the trajectories' mean and standard deviations. Our approach led to some non-intuitive findings. MCTS training generally requires a large number of samples and simulations. However, we observed that bootstrapping tree search with a pre-trained policy allows us to achieve high quality results with a low MCTS branching factor and few simulations. Without the proposed policy bootstrapping, continuous MCTS would require a much larger branching factor and simulation count, rendering it prohibitively expensive. In our experiments, we use PPO as our baseline policy optimization algorithm. TPO significantly improves the policy on nearly all the environments. For example, in complex environments such as Humanoid with a 17 dimensional action space, we achieve a $2.5\times$ improvement over the baseline algorithm.

## 1 Introduction

Fueled by advances in neural representation learning, the field of model-free reinforcement learning has rapidly evolved over the past few years. These advances are due in part to the advent of algorithms capable of navigating larger action spaces and longer time horizons [2, 32, 42], as well as the distribution of data collection and training across massive-scale computing resources [42, 38, 16, 32].

While learning algorithms have been continuously improving, it is undeniable that tree search methods have played a large role in some of the most successful applications of RL (e.g., AlphaZero [42], Mario [10] and Arcade games [48]). Tree search methods enable powerful explorations of the action space in a way which is guided by the topology of the search space, focusing on branches (actions) that are more promising. Although tree search methods have achieved impressive results on a range of discrete domains, they have not yet fulfilled their true potential in continuous domains. Given that the number of actions is inherently unbounded in continuous domains, traditional approaches to building the search tree become intractable from a computational perspective.

In this paper, we introduce **TPO**, a **T**ree Search **P**olicy **O**ptimization for environments with continuous action spaces. We address the challenges of building the tree and running simulations by adopting a hybrid method, in which we first train a policy using existing model-free RL methods, and then use the pre-trained policy distribution to draw actions with which to build the tree. Once the tree has been constructed, we run simulations to generate experiences using an Upper Confidence Bounds for Trees (UCT) approach [33]. Populating the tree with the action samples drawn from a pre-trained policy enables us to perform a computationally feasible search. TPO is a variation of the policy iteration method [35, 44, 42, 1]. Broadly, in these methods, the behavior of policy is iteratively updated using the trajectories generated by an expert policy. Then, the newly updated policy in return guides the expert to generate higher quality samples. In TPO, we use tree search as an expert to generate high quality trajectories. Later, we employ the updated policy to re-populate the tree search. For tree search, we use the Monte Carlo Tree Search (MCTS) [5] expansion and selection methods. However, it is

challenging to directly infer the probability of selected actions for rollout; unlike in discrete domains where all actions can be exhaustively explored, in continuous domains, we cannot sample more than a subset of the effectively innumerable continuous action space. Furthermore, to use the trajectories generated by MCTS, we must perform off-policy optimization. To address this challenge, we define a new loss function that uses the weighted mean and standard deviation of the tree search statistics to update the pre-trained policy. For ease of implementation and scalability, we use Proximal Policy Optimization (PPO) [38] and choose as our policy optimization baseline.

Our approach led to some non-intuitive findings. MCTS training generally requires a large number of branches and simulations. For example, AlphaGo uses 1600 simulations per tree search and a branching factor of up to 362 [40]. However, we observed that if we pre-train the policy, we require far fewer simulations to generate high quality trajectories. While we do benefit from exploring a greater number of branches, especially for higher dimensional action spaces (e.g. Humanoid), we observed diminishing returns after only a small number of branches (e.g., 32) across all of the evaluated environments. Furthermore, performance quickly plateaued as we increased the number of simulations past 32. This property did not hold when we initialized tree search with

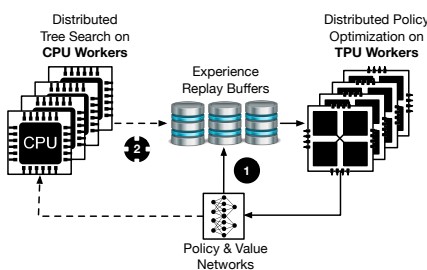

**Figure 1:** **T**ree Search **P**olicy **O**ptimization (**TPO**) Framework. In phase 1, we perform a policy gradient based optimization training to build a target policy. In phase 2, we iteratively build an MCTS tree using the pre-trained target policy and update the target policy using roll-out trajectories from MCTS. Both training and data collection are done in a distributed manner.

an untrained policy. This is a critical advantage of our method as it would otherwise be computationally infeasible to generate high quality trajectories using tree search. The main contributions of TPO are summarized as follows:

1. **Tree search policy optimization for continuous action spaces**. TPO is one of the very first techniques that integrates tree search into policy optimization for continuous action spaces. This unique integration of tree search into policy optimization yields a superior performance compared to baseline policy optimization techniques for continuous action spaces.

2. **Policy bootstrapping.** We propose a policy bootstrapping technique that significantly improves the sample efficiency of the tree search and enables us to discretize continuous action spaces into only a few number of highly probable actions. More specifically, TPO only performs 32 tree searches compared to substantially larger number of tree searches (1600, $50\times$ more) in AlphaGo [42]. In addition, TPO narrows down the number of tree expansion (actions) compared to discretization techniques such as Tang et al. [45] which requires 7-11 bins per action dimension. This number of bins translates to a prohibitively large number of actions even in discrete domain for complex environments such as Humanoid which has a 17 dimensional action space. In contrast, TPO only samples 32 actions at each simulation step across all the environments.

3. **Infrastructure and results.** On the infrastructure side, we developed a distributed system (shown in Figure 1), in which both policy optimization and data collection are performed on separate distributed platforms. The policy optimization is done on a TPU-v2 using multiple cores, and MCTS search is performed on a rack of CPU nodes. A synchronous policy update and data collection approach is used to train the policy and generate trajectories. TPO readily extends to challenging and high-dimensional tasks, such the Humanoid benchmark [9]. Our empirical results indicate that TPO significantly improves the performance of the baseline policy optimization algorithm, achieving up to $2.5\times$ improvement.

## 2 BACKGROUND

Policy iterations methods have shown promising results in reinforcement learning. AlphaZero [42], a major breakthrough in reinforcement learning, uses MCTS as an expert to train a discrete policy model for the game of Go. EXiT [1] established theoretical grounds for policy iterations using MCTS in discrete domains. A recent work [28] proposes some theoretical extensions to AlphaZero's MCTS method [42] for environments with continuous action spaces. However, the authors only evaluate their method on the Pendulum OpenAI Gym environment which has only a one dimensional action

space. It is not obvious how the approach can scale to more complex environments with much higher dimensions. While TPO also belongs to the category of policy iteration methods, it studies a rather less explored territory of using MCTS as expert for complex environments with continuous action spaces.

Policy optimization falls into two major categories: on-policy and off-policy optimization. In on-policy methods, the policy is optimized using samples generated from the current policy, whereas with off-policy algorithms, we can also train the policy using experiences from other sources, such as an expert. Although on-policy reinforcement learning methods, such as TRPO [36], PPO [38], PGQL [31], and A3C [27] have been extremely successful, they are known to be sample inefficient [16, 12], as they must generate fresh trajectories between each gradient update and cannot reuse these trajectories. This effect becomes more pronounced as the complexity of the task grows, requiring millions of steps to learn a policy. On the other hand, off-policy methods (e.g., DQN [26, 27], DDPG [24]) are more sample efficient, as each example is used multiple times for training. However, these methods require more extensive hyper-parameter tuning, suffering from instability and slow convergence [30, 18, 16, 34, 47].

A few recent methods have been developed to embed differentiable planning procedures in computation graphs [15, 43, 25, 14, 19]. MCTSnets [15] is a deep learning model that incorporates simulation-based search with a vector embedding for expansion, evaluation, and backup. The three policies are jointly optimized with an end-to-end gradient based approach. The authors only report results on the classic planning benchmark, Sokoban, which is a simple discrete optimization environment. Extension of this approach to continuous domains is non-obvious. SoRB [11] is another approach that combines planning with policy optimization. However, the target problem space of SoRB is only *goal-reaching* tasks. Although there has been previous work on continuous MCTS, none has been conducted in the context of deep reinforcement learning [8, 13, 7]. More sophisticated approaches to node expansion improve the quality of search [7], but TPO only uses a simple scheme in which a fixed number of actions are sampled from the policy. We found our approach both simple to implement and effective, without the need for extensive fine-tuning or hyper-parameter optimization. For example, we found that across the evaluated OpenAI Gym environments, a sample size of 32 yields strong results. In our experiments, we chose PPO [38], which has been shown to yield superior performance in various large-scale problems such as Hide and Seek [2] and Dota [32], to pre-train the policy used to populate the MCTS tree. However, our framework is independent of the specific choice of baseline policy and can be readily applied to other model-free RL algorithms, such as SAC [16] or TD3 [34].

## 3  MONTE CARLO TREE SEARCH FOR CONTINUOUS ACTION SPACES

**Preliminaries and notations.** Policy optimization in continuous environments can be well modeled by a Markov Decision Process (MDP) [3], defined by the tuple $(\mathcal{S}, \mathcal{A}, p, r, \gamma)$, where $\mathcal{S}(\mathcal{A})$ is a set of states (actions) in continuous space. The state transition probability $p(\mathbf{s}_{t+1}, \mathbf{s}_t, \mathbf{a}_k)$ represents the probability desnsity of observing the next state in time, $\mathbf{s}_{t+1} \in \mathcal{S}$, given the current $\mathbf{s}_t \in \mathcal{S}$ and the current action $\mathbf{a}_k \in \mathcal{A}$. Taking action $\mathbf{a}_k$ at state $\mathbf{s}_t$, the environment moves to a new state $\mathbf{s}_{t+1}$ and returns a bounded reward $r(\mathbf{s}_t, \mathbf{a}_k) : \mathcal{S} \times \mathcal{A} \to [r_{min}, r_{max}]$. Finally, $\gamma \in (0, 1)$ represents the discount factor, used to calculate the discounted cumulative expected reward. We use $\pi_\theta(\mathbf{a}_k | \mathbf{s}_t)$ to represent the neural network policy with trainable parameters $\theta$. We use $v_\phi(\mathbf{s}_t)$ to represent the value from a neural network with parameters $\phi$.

### 3.1  MONTE CARLO TREE SEARCH ALGORITHM FOR CONTINUOUS ACTION SPACE

Previous work on Monte Carlo Tree Search (MCTS) was mostly focused on zero-sum games with discrete action spaces [42, 41, 29, 40, 5, 8, 22, 39]. In this section, we will highlight our adaptation of MCTS for environments with continuous action spaces and non-zero immediate rewards. For the complete definition of MCTS, we invite the readers to see [42]. To make MCTS possible, we assume the environment can be set to an arbitrary state. This assumption can be relaxed using a learned model of the environment (See §6).

In this work, we use a variant of Polynomial Upper Confident Tree (PUCT) algorithm in TPO for MCTS [33]. Nodes of the Monte Carlo tree are states of the environment $\mathbf{s}_t \in \mathcal{S}$ and edges are actions, $\mathbf{a}_k \in \mathcal{A}$. We store a set of statistics for each edge of Monte Carlo tree: the number of visits $N$, the cumulative sum of action value $W$, and the average of the action value $Q$. We update these statistics after each MCTS simulation for the traversed edges. Each MCTS simulation starts from a root state

and consists of four steps: select, expand, evaluate, and backup. Here we only overview the select and expand steps, as they are the contributions of TPO to enable tree search for continuous action spaces.

**❶ Select.** We follow a variant of Polynomial Upper Confident Tree (PUCT) algorithm [33, 42] to select actions. For each node, $\mathbf{s}_t$, we select an action: $\mathbf{a} = \operatorname{argmax}_{\mathbf{a}_k} Q(\mathbf{s}_t, \mathbf{a}_k) + U(\mathbf{s}_t, \mathbf{a}_k)$, where $U(\mathbf{s}_t, \mathbf{a}_k)$ is:

$$U(\mathbf{s}_t, \mathbf{a}_k) = c_{puct} \cdot \pi_\theta(\mathbf{a}_k | \mathbf{s}_t) \cdot \frac{\sqrt{N(\mathbf{s}_t)}}{1 + N(\mathbf{s}_t, \mathbf{a}_k)} \tag{1}$$

where $c_{puct}$ is a constant exploration coefficient and $\pi_\theta(\mathbf{a}_k | \mathbf{s}_t)$ represents the prior probability assigned to each action by the policy network. To further promote exploration, we add a Dirichlet noise, $\mathcal{D}ir(\alpha)$[1], to the prior probabilities of the actions at the root node [42].

To support environments with immediate rewards, we propose to use the following equation for estimating the value of an action:

$$Q(\mathbf{s}_t, \mathbf{a}_k) = r(\mathbf{s}_t, \mathbf{a}_k) + \gamma \frac{W(\mathbf{s}_t, \mathbf{a}_k)}{N(\mathbf{s}_t, \mathbf{a}_k)} \tag{2}$$

where $W(\mathbf{s}_t, \mathbf{a}_k)$ is the cumulative sum over the backed up values from its children.

**❷ Expand.** In discrete environments such as the game of Go [42] or Chess/Shogi [41], all possible actions are considered in MCTS expansion. However, in environments with continuous action spaces, the set of possible actions is innumerable. Hence, enumerating all the possible actions for each node is impossible. To circumvent this challenge, we propose to draw samples from the policy distribution. These sampled actions will become the edges of the leaf node.

After a fixed number of simulations, we select an action with the largest visit counts from the current root node and advances the environment. We use $Q(\mathbf{s}_t, \mathbf{a}_k) + U(\mathbf{s}_t, \mathbf{a}_k)$ as a tie-breaker between the actions with the same number of visit counts. We also define $\hat{\mu}_{TS}(\mathbf{s}_t)$ as an unbiased estimate [46, 27] for the mean value of the tree policy at a given node computed across each action space dimension:

$$\hat{\mu}_{TS}(\mathbf{s}_t) = \sum_k \frac{N(\mathbf{s}_t, \mathbf{a}_k)}{N(\mathbf{s}_t)} \cdot \mathbf{a}_k \tag{3}$$

Later in Section 4, we use this unbiased mean estimate to optimize the policy distribution. All MCTS trajectories are added to a replay buffer, which is then used to optimize the policy and value networks.

## 4 TREE SEARCH POLICY OPTIMIZATION (TPO)

**Neural network architecture.** Both value, $v_\phi(.)$, and policy networks, $\pi_\theta(.)$, are represented by a fully-connected multi-layer perceptron (MLP) with two hidden layers of 64 units. The output of each layer is passed through a $\tanh(\cdot)$ non-linearity function. The weights of value and policy networks are not shared. The value network outputs a single scalar value $v(\mathbf{s}_t)$, whereas the policy network outputs a vector of means $\boldsymbol{\mu}(\mathbf{s}_t)$, and variances $\boldsymbol{\sigma}(\mathbf{s}_t)$ for a number of independent Normal distributions for each action dimension.

**Loss functions.** We first describe the loss functions for the on-policy phase and then delve into the loss functions of the proposed off-policy tree search optimization. For ease of implementation, we used PPO [38] for on-policy optimization. The policy loss function of PPO is:

$$L^{\text{CLIP}}(\theta) = \hat{\mathbb{E}}_t[\max(-\mathfrak{r}_t(\theta) \cdot \hat{A}_t, -\text{clip}(\mathfrak{r}_t(\theta), 1 - \epsilon, 1 + \epsilon) \cdot \hat{A}_t)]; \quad \text{where} \quad \mathfrak{r}_t(\theta) = \frac{\pi_\theta(\mathbf{a}_t | \mathbf{s}_t)}{\pi_{\theta_{old}}(\mathbf{a}_t | \mathbf{s}_t)} \tag{4}$$

where $\mathfrak{r}_t(\theta)$ is the probability ratio between the new policy, $\pi_{\theta_t}(\mathbf{a}_k | \mathbf{s}_t)$, and the old policy, $\pi_{\theta_{old}}(\mathbf{a}_k | \mathbf{s}_t)$, PPO clips the probability ratio to be within the range $[1 - \epsilon, 1 + \epsilon]$, which penalizes large updates to the policy. $\hat{A}_t$ is the truncated generalized advantage estimate [38, 37], computed for a trajectory of length $T$. To update the value network, we use a variant of PPO[2] that also applies clipping to the value function:

$$V_\phi^{\text{CLIP}}(\mathbf{s}_t) = V_{\phi_{old}}(\mathbf{s}_t) + \text{clip}(V_\phi(\mathbf{s}_t) - V_{\phi_{old}}(\mathbf{s}_t), -\epsilon, \epsilon) \tag{5}$$

---

[1] $\mathcal{D}ir(\alpha)$ returns a noise vector of $\{\eta_1, \eta_2, ..., \eta_n\}$ where $\sum_{i=1}^n \eta_i = 1$. $\alpha \in \mathbb{R}^+$ is Dirichlet noise scaling parameter that controls the shape of the generated noise vector.

[2] https://github.com/openai/baselines/tree/master/baselines/ppo2

---

**Algorithm 1: TPO**: **T**ree Search **P**olicy **O**ptimization

---

1  **for** iteration ← 1, 2, ..., number of iterations **do**
2      **if** iteration ≤ $\rho$ × number of iterations                                                    ▷ I ∈ (0, 1)
3      **then**
4          $\pi_c \leftarrow \pi_{(\theta_{old})}$
5          $L_c \leftarrow L_{\text{On-}\pi}^{\text{CLIP+VF}}(\theta;\phi)$                          ▷ On-Policy Training; See Eq. 4, 5, 6, and 7
6      **else**
7          $\pi_c \leftarrow \pi_{(\theta_{old};TS)}$
8          $L_c \leftarrow L_{\text{Off-}\pi}^{\text{PF+VF}}(\theta;\phi)$                       ▷ Off-Policy Training; See Eq. 5, 6, 8, and 9
9      Roll out trajectories $\{\tau_1, \tau_2, ..., \tau_\kappa\}$ for T timesteps each using $\pi_c$
10     **for** e ← 1, 2, ..., number of epochs **do**
11         Optimize $L_c$ wrt to $\theta$ and $\phi$ with minibatch size M
12     $\theta_{old} \leftarrow \theta$
13     $\phi_{old} \leftarrow \phi$

---

**Table 1:** The benchmark suite (six OpenAI Gym [4] environments) with their corresponding observation/action spaces.

| Environments | State Space | Action Space |
|---|---|---|
| Ant-v2 | $\mathbb{R}^{111}$ | $[-1.0, 1.0]^8$ |
| HalfCheetah-v2 | $\mathbb{R}^{17}$ | $[-1.0, 1.0]^6$ |
| Hopper-v2 | $\mathbb{R}^{11}$ | $[-1.0, 1.0]^3$ |
| Humanoid-v2 | $\mathbb{R}^{376}$ | $[-0.4, 0.4]^{17}$ |
| Swimmer-v2 | $\mathbb{R}^8$ | $[-1.0, 1.0]^2$ |
| Walker2d-v2 | $\mathbb{R}^{17}$ | $[-1.0, 1.0]^6$ |

$$L^{\text{VF}}(\phi) = \hat{\mathbb{E}}_t[\max((V_\phi^{\text{CLIP}}(\mathbf{s}_t) - V_t^{targ})^2, (V_\phi(\mathbf{s}_t) - V_t^{targ})^2)] \tag{6}$$

Finally, the total loss combines the policy surrogate and value function terms with a regularization coefficients $c_1$:

$$L_{\text{On-}\pi}^{\text{CLIP+VF}}(\theta;\phi) = L^{\text{CLIP}}(\theta) + c_1 \cdot L^{\text{VF}}(\phi) \tag{7}$$

In the off-policy phase, we use the same value function as PPO for the value updates, as shown in Eq. 5 and Eq. 6. For the policy updates, we define a new loss function, derived from the collected MCTS statistics:

$$\text{L}^{\text{PF}}(\theta) = \hat{\mathbb{E}}_t[(\hat{\mu}_{\text{TS}}(\mathbf{s}_t) - \mu_{\pi_\theta}(\mathbf{s}_t))^2 + c_2 \cdot (\hat{\sigma}_{\text{TS}}(\mathbf{s}_t) - \sigma_{\pi_\theta}(\mathbf{s}_t))^2] \tag{8}$$

$L^{\text{PF}}(\theta)$ is the sum of the squared-error loss between the unbiased mean estimate and the standard deviation estimate, calculated from the MCTS trajectories, and the mean and standard deviation of the policy distribution at a given state $\mathbf{s}_t$. The intuition for choosing this loss term for policy updates is to incentivize moving the mean of the policy distribution towards an unbiased estimate computed from the tree search. Finally, we combine the defined TPO policy loss and value loss to obtain the following objective, which is maximized at each training step:

$$L_{\text{Off-}\pi}^{\text{PF+VF}}(\theta;\phi) = L^{\text{PF}}(\theta) + c_3 \cdot L^{\text{VF}}(\phi) \tag{9}$$

If the policy and value networks do not share parameters, the regularization coefficients $c_1$ and $c_3$ become irrelevant. In neither $L_{\text{On-}\pi}$ or $L_{\text{Off-}\pi}$ losses, we do *not* use any entropy bonus term.

**Algorithm.** Algorithm 1 fleshes out the proposed TPO algorithm, which uses a combination of policy gradient and MCTS. In the first phase of training, we use the policy network to roll out trajectories and optimize the PPO loss function (See Eq. 7). In the second phase, we switch to an off-policy mode, in which we only use trajectories from MCTS to optimize the policy. That is, we construct the loss (See Eq. 9.) on the roll out trajectories generated by the tree search. We treat $\rho$, which defines the switching ratio between policy gradient and tree search, as a hyperparameter of the TPO algorithm. The intuition behind bootstrapping tree search with a pre-trained policy is that continuous action spaces require a relatively stable policy to encourage the exploration of trajectories with higher returns, and an untrained policy cannot adequately guide the tree search. Our experiments support this hypothesis; if we start from a lightly trained policy (i.e. small switching ratio, $\rho$), the final performance is severely diminished. The same reasoning applies to choosing to start with policy gradient optimization in the first phase of training followed by tree search policy optimization.

## 5 EXPERIMENTS

We evaluate TPO on a variety of continuous environments from OpenAI Gym [4] (Table 1). Table 2 show the hyperparameters used for training the value and policy networks and those used for tree search. Our policy training is done on a TPU-v2 platform with eight cores, and MCTS is performed on CPU. We chose PPO [38] for the on-policy optimization update. For each environment, we run TPO for four million time steps to estimate the gradients.

To efficiently perform MCTS sampling for trajectory rollouts, we use BatchEnv [17] that extends the OpenAI Gym [4] interface to multiple parallel environments. Using BatchEnv, we evaluate up to 32 actions in parallel and obtain their associated rewards. We use 32 Intel CPUs, each assigned to one environment instance, in order to parallelize simulation. We found that 32 is a reasonable choice for the maximum number of MCTS simulations per each action selection, as it strikes a good balance between search latency and policy performance. Using our parallel infrastructure, the latency of action selection is somewhere between 0.2-0.3 sec when we use 32 MCTS simulations per action.

Unless stated otherwise, in all of the experiments, we report the average return of the last 100 episodes across five runs (each with a different random seed). To make a fair comparison with the baseline algorithm (PPO), we reproduce the reported results in [38] using our infrastructure. In all the environments, we use the best obtained results for PPO using our infrastructure, which in some environments (HalfCheetah-v2 and Walker2d-v2) outperform the reported results in [38]. In all the experiments, we use Adam [21] as our optimizer and run training for 10 epochs per policy update.

**Summary of results.** Table 3 depicts the performance comparison between TPO and PPO across six environments with continuous action spaces. In all the environments, TPO yields a better total return compared to PPO, generally by a significant margin. Using TPO, we observe a maximum improvement of roughly 2,900 in Ant-v2 and a minimum improvement of only 7 in Swimmer-v2. Swimmer-v2 is the simplest environment in the set of evaluated benchmarks with an action space of only two dimensions. On the other hand, Ant-v2 is a much more complex benchmark with an action-space of eight dimensions. The delta in performance improvement suggests that TPO is better able to leverage tree search exploration to improve policies in complex environments with large action spaces. Figure 2 shows the training progress of TPO and PPO across six environments. To conduct this experiment, we start by pre-training with PPO. At $\sim$ 2M time steps, TPO restores checkpoints from PPO training and continues training using tree search. As such, the total return curve for both TPO and PPO overlap before reaching 2M time steps. In all the environments, we observe a sharp spike in the total return curve after switching to tree search, which can be attributed to the effectiveness of the proposed tree search method for policy optimization. For complex environments, such as Humanoid-v2, we observe continuing improvement after switching to MCTS-based policy optimization. Ablation Studies To better understand the impact of key components of TPO, we perform a set of ablation studies. Concretely, we show the results for

**Table 2:** TPO hyperparameters.

| MCTS Hyperparameters | Value |
|---|---|
| Number of MCTS simulations | 32 |
| Maximum number of actions per node | 32 |
| $c_{puct}$ | 3 |
| Temperature ($\tau$) | 5 |
| Dirichlet noise scaling parameter ($\alpha$) | 1.75 |
| Dirichlet noise weight ($\omega$) | 0.5 |
| Training Hyperparameters | Value |
| Policy network architecture | FC:64 – FC:64 |
| Value network architecture | FC:64 – FC:64 |
| Adam learning rate ($\eta$) | $3 \times 10^{-4}$ and $9 \times 10^{-4}$ |
| Horizon (T) | 2048 |
| Switching ratio ($\rho$) | 0.5 |
| Clipping coefficient ($\epsilon$) | 0.2 |
| Number of epochs | 10 |
| Minibatch size (M) | 128 |
| Gradient norm clipping coefficient | 0.5 |
| Discount ($\gamma$) | 0.99 |
| GAE parameter ($\lambda$) | 0.95 |

**Table 3:** Performance comparison between TPO and PPO after 4M time-steps. $\Delta$ values show the difference between TPO and PPO for each category. A positive $\Delta$ value indicates that TPO outperforms PPO.

| Benchmarks | Min. | | | Avg. | | | Max. | | |
|---|---|---|---|---|---|---|---|---|---|
| | PPO | TPO | $\Delta$ | PPO | TPO | $\Delta$ | PPO | TPO | $\Delta$ |
| Ant-v2 | 1722.75 | 4455.57 | +2732.81 | 1873.85 | 4777.06 | +2903.21 | 2022.08 | 5011.08 | +2989.0 |
| HalfCheetah-v2 | 1658.70 | 1788.77 | +130.06 | 2530.81 | 3949.05 | +1418.24 | 5049.07 | 6734.45 | +1685.38 |
| Hopper-v2 | 1744.24 | 2205.47 | +461.22 | 1937.82 | 3196.14 | +1258.32 | 2051.89 | 3613.36 | +1561.47 |
| Humanoid-v2 | 4269.44 | 5391.43 | +1121.99 | 4439.58 | 5460.93 | +1021.35 | 4609.72 | 5530.43 | +920.71 |
| Swimmer-v2 | 109.91 | 119.48 | +9.57 | 113.74 | 120.81 | +7.07 | 116.74 | 122.39 | +5.65 |
| Walker2d-v2 | 3605.27 | 3788.83 | +183.56 | 3942.57 | 4092.21 | +149.63 | 4155.37 | 4667.64 | +512.27 |

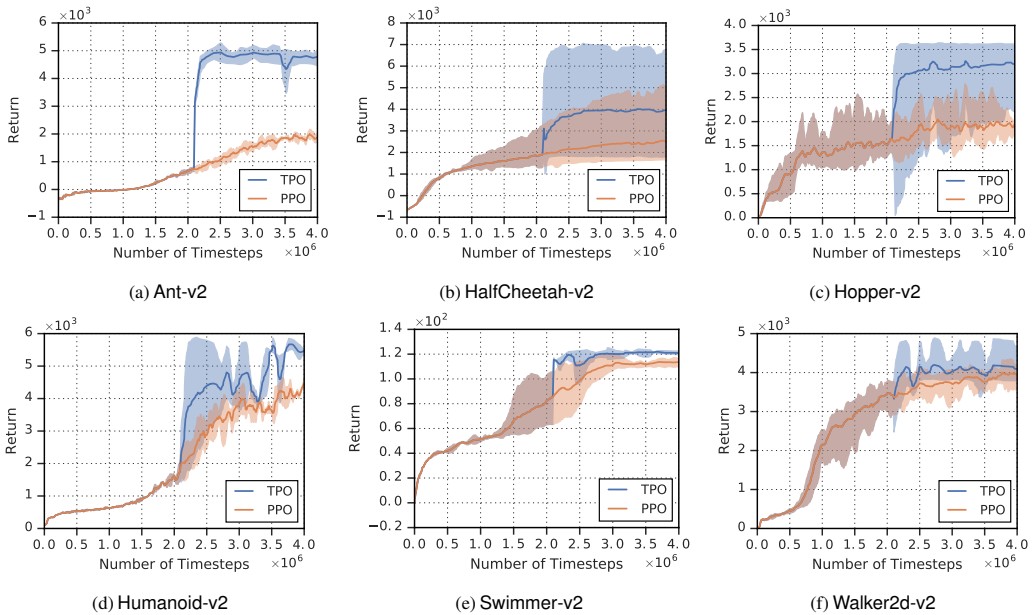

**Figure 2:** Comparison between PPO and TPO training curves. Each data point represents the average return of the last 100 training episodes. MCTS rollout starts after 50% of the total number of timesteps, i.e. ∼2M.

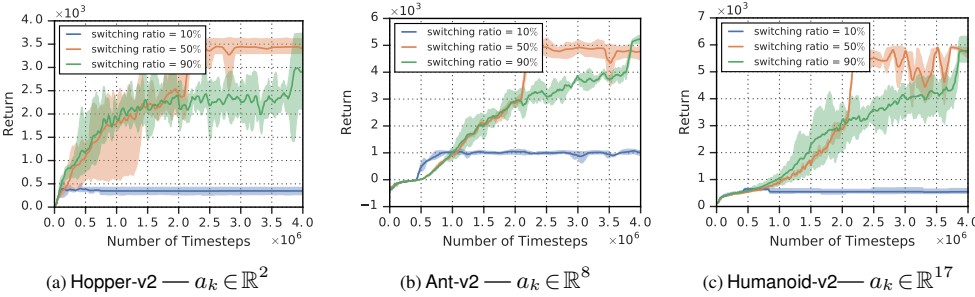

**Figure 3:** TPO training curves when the switching ratio (ρ in Algorithm 1) is 10%, 50% (default value in TPO), and 90%.

three environments with different levels of action space complexity, namely Hopper-v2, Ant-v2, and Humanoid-v2 with action spaces of three, eight, and 17, respectively.

**Switching ratio.** First, we study the impact of varying the switching ratio (ρ in Algorithm 1) between 10%, 50% (default value in TPO), and 90% across three tasks. As shown in Figure 3, prematurely switching ($\rho = 10\%$, blue line in Figure 3) to tree search optimization severely harms policy performance, showing no further improvement after switching to MCTS optimization. This behavior can be attributed to the fact that in tree search policy optimization for environments with continuous action spaces, it is of utmost importance to have a reasonably well-trained policy to guide the tree search toward trajectories with potentially higher return values. Switching to tree search at the very end of training ($\rho = 10\%$, green line in Figure 3) can be damaging as well. While switching to tree search at the 90% of the total training steps in Hopper-v2 does not improve the average return compared to switching after 50% of the total training steps, it slightly improves the average return in Ant-v2. That is, the point at which to switch to tree search is a hyperparameter that should be adjusted carefully for better policy performance.

**Number of actions per node.** Figure 4 shows the impact of varying the number of actions per node in tree search between 4, 8, and 32 (default in TPO is 32). This study reveals the following findings. (1) variance reduction in return values; increasing the number of actions per node significantly reduces the variance of the total return value (i.e. shrinkage in the shaded area), which further improves the robustness of the training process. (2) better policy performance (higher return value); while the impact of increasing the number of actions per node is marginal in simple environments, such as Hopper-v2 and Ant-v2, exploring more actions per node in environments with a more complex action

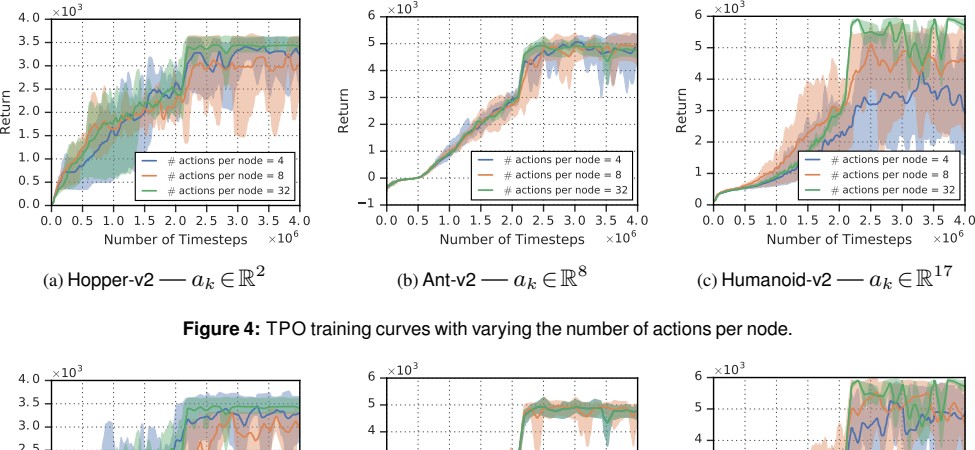

**Figure 4:** TPO training curves with varying the number of actions per node.

**Figure 5:** TPO training curves with varying number of MCTS simulations.

space, Humanoid-v2, has significant benefit for policy performance. As shown in Figure 4c, changing the number of actions per node from eight to 32 increases the average total return by ∼1100.

**Number of MCTS simulations.** Figure 5 shows the impact of the number of MCTS simulations (See Section 3) on the performance of the policy. While increasing the number of MCTS simulations generally improves performance, we found that increasing the branching factor (number of actions per node) had more impact. For example, for Humanoid-v2, the performance gap between 8 and 32 simulations (shown in Figure 5c), is less pronounced than the improvement from increasing the number of actions per node from 8 and 32 (shown in Figure 4c). These studies suggest that it is more beneficial to devote the tree search budget to exploring more actions per node, especially for more complex environments, rather than conducting a larger number of tree search simulations.

## 6    DISCUSSION

In this paper, we have studied Monte Carlo Tree Search in continuous space for improving the performance of a baseline on-policy algorithm [38]. Our results show that MCTS policy optimization can indeed improve the quality of policy in choosing better actions during policy evaluation at the cost of more samples during MCTS rollout. We show that bootstrapping tree search with a pre-trained policy enables us to achieve high performance with a low MCTS branching factor and few simulations. On the other hand, without pre-training, we require a much larger branching factor and simulation count, rendering MCTS computationally infeasible. One of the future research direction is to explore techniques for improving the sample efficiency and removing the need for having a reset-able environment. To achieve these goals, we can use a trained model of the environment similar to model-based reinforcement learning approaches [20, 23, 6], instead of interacting directly with the environment in MCTS. Recently, MBPO [20] showed that they can train a model of Mujoco environments that is accurate enough for nearly 200-step rollouts in terms of accumulated rewards. This level of accuracy horizon is more than enough for the shallow MCTS simulations (32 simulations) that is employed in TPO. As mentioned earlier, TPO assumes access to an environment that can be restarted from an arbitrary state for MCTS simulations. While this assumption can be readily satisfied in some RL problems such as playing games, it may be harder to achieve for physical RL problems like Robotics. This assumption can also be relaxed using a modeled environment to replace the interactions with the real environment during MCTS simulations.

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
