# OpenReview forum: "TPO: TREE SEARCH POLICY OPTIMIZATION FOR CONTINUOUS ACTION SPACES"
_ICLR.cc/2020/Conference — Reject_

### Official Review · AnonReviewer2 · 2019-10-23
**Official Blind Review #2**

**Rating:** 3

**Review:**


# Summary
The paper proposes to use MCTS for fine-tuning a policy in continuous control tasks. Action selection is done by PUCT [42] and node expansion is done by sampling from the policy instead of trying all actions. Furthermore, the importance of good parallel implementation is highlighted. Results on MuJoCo tasks show a bit of a gain in performance but with quite high variance.

# Decision
There are some concerns regarding the novelty of the proposed method. Furthermore, evaluations seem rather noisy to make a reliable judgement. Therefore, I currently refrain from recommending this paper for publication.

# Discussion
1) Being not an expert on MCTS, it nevertheless appears to me that it is a natural idea to extend it to continuous problems. A quick search reveals a number of papers, e.g.,

https://ieeexplore.ieee.org/document/8401544
https://tel.archives-ouvertes.fr/tel-00927252/document
https://openreview.net/forum?id=SyiF5-23Z
http://proceedings.mlr.press/v80/lee18b/lee18b.pdf

=> Can you comment on what exactly is your contribution?

2) Experiments
    - I am not sure that showing the burn-in period of pure PPO learning in Fig. 2 is informative. Clearly, if one starts with a good policy, it can only get better from there. So, one can directly start with a PPO-pretrained policy.
    - For the same reason, the switching ratio experiments in Fig. 3 seem superfluous.
    - In MuJoCo experiments in Fig. 2, in the top-right two plots, there is quite high variance; in the bottom plots, on the other hand, the variance grows and shrinks. How can you explain such oscillating behavior of the variance among runs? How many trials were performed? The curves do not seem to be reliable.
    - Evaluations in Figs. 4–5 are quite indecisive, except for Humanoid maybe. Perhaps, more representative environments could be chosen.
    - The algorithm is not compared to any other approach. Are there no similar methods?


**Experience Assessment:**

I have published one or two papers in this area.

**Review Assessment: Checking Correctness Of Derivations And Theory:**

I assessed the sensibility of the derivations and theory.

**Review Assessment: Checking Correctness Of Experiments:**

I assessed the sensibility of the experiments.

**Review Assessment: Thoroughness In Paper Reading:**

I read the paper at least twice and used my best judgement in assessing the paper.

---

> ### Author Response · Authors · 2019-11-15
> **Response to Reviewer #2**
>
>
> == Switching Ratio Experiment ==
>
> The switching ratio experiment highlights the trade-off between the number of MCTS simulations and the quality of the results. A fully-trained policy (with low entropy) prevents MCTS from exploring the action space and focuses instead on maximizing reward (only exploitation). On the other end of the spectrum, since MCTS search relies on guidance from the policy for action selection, starting from a premature policy misleads MCTS search. In this situation, one approach would be to increase the number of branches in MCTS search which significantly increases the MCTS simulation time.
>
> == Figure 2 ==
>
> One main reason for observing a high variance at the switching point (switch from pure policy iteration to MCTS) is due to the fact that the collected data from MCTS at the beginning has high variance from the pre-trained policy. But as we move forward, the variance reduces which is an indication of the fact that policy is learning the distribution of MCTS. In general, after a couple of training iterations, the variance of results between PPO and TPO follows the same trend. Note that, even though the variance of results between PPO and TPO are nearly the same, the absolute value of the results from TPO is much higher than PPO in multiple environments (Ant, HalfCheetah, Hopper, and Humanoid).
>
> == Figure 4 and 5 ==
>
> The main purpose of these experiments is to show the impact of sampling of actions per node (state) and number of MCTS simulations. The results highlight two main outcome of our approach: (1) improving the quality of results (e.g. Humanoid) and (2) reducing the variance (e.g. Hopper and Ant).

---

### Official Review · AnonReviewer1 · 2019-10-25
**Official Blind Review #2**

**Rating:** 3

**Review:**

This paper proposes a hybrid approach that combines MCTS with policy optimization. The main idea is to use PO for policy improvement, then boostrapping for action selection. The proposed hybrid framework enables MCTS planning on continuous action problems. The method is demonstrated on continuous control tasks such as Humanoid with high-dimensional and continuous actions. Integrating planning into PO like this way is shown a large improvement over baseline methods.

Overall, this paper pursues an interesting research problem. Integrating a planning ability to policy optimization would definitely be desired to achieve more data-efficiency. The idea proposed in this paper is quite straightforward. It simply proposed to use policy gradient algorithms to optimize the rolout policy and the action selection policy. In addition, some of the descriptions in the paper are unclear that makes it hard to understand. My main concerns are as follows.

- The policy network is trained using a gradient-based approach. However, the original MCTS framework is expected to optimize a globally optimal policy. It would be more valuable if the authors pay some discussions for this limitation and demonstrate it in experiments. Probably if it is evaluated on a discrete domain first.

- As said this hybrid approach not only aims to extend MCTS to continuous domains but could also be considered as a way of integrating a planning ability into policy optimization that helps data-efficiency. Therefore more related work and discussions, and drawing connections with planning embedding and model-based policy optimization are very helpful.

- There are many technical details missing, hence making it difficult to understand: i) how a search tree can be built for continuous domains?, I do not see how the branching factor and the policy boostrapping are used to simplify a search tree construction for continuous problems; ii) how the description in section 3 is related to the proposed framework? iii) what is the meaning of the Number of actions per node, it is unclear which nodes? How the tree is constructed based on this number of nodes?

- Experimental results are quite promising as expected. I wonder how TPO and PPO are compared in terms of the total computation time?

**Experience Assessment:**

I have published one or two papers in this area.

**Review Assessment: Checking Correctness Of Derivations And Theory:**

I assessed the sensibility of the derivations and theory.

**Review Assessment: Checking Correctness Of Experiments:**

I assessed the sensibility of the experiments.

**Review Assessment: Thoroughness In Paper Reading:**

I read the paper at least twice and used my best judgement in assessing the paper.

---

> ### Author Response · Authors · 2019-11-15
> **Response to Reviewer #2**
>
> Thank you for recognizing our promising results.
>
> == Comparison with other MCTS Approaches ==
>
> Thank you for bringing these papers to our attention. We will make sure to cite them in the related work.
>
> “Monte-Carlo Tree Search vs. Model-Predictive Controller: A Lane-Following Example”
> “Monte Carlo Tree Search for Continuous and Stochastic Sequential Decision Making Problems”
>
> These papers proposed to use MCTS solely for planning, whereas our main contribution is the seamless integration of MCTS-guided planning and policy optimization.
>
> In “Deep Reinforcement Learning in Continuous Action Spaces: a Case Study in the Game of Simulated Curling”, the authors do not explore the implications of having environments with immediate reward in MCTS and followed a similar environment setting as AlphaZero. As such, it makes it hard to have a direct comparison with their approach.
>
> In “Continuous Control Monte Carlo Tree Search Informed by Multiple Experts”, as mentioned in the limitations, the authors explored incorporating MCTS search in a supervised setting, as opposed to using MCTS samples in a reinforcement learning setting, particularly for policy iteration methods.
>
> == MCTS Global Optimization vs. Local Optimization ==
>
> AlphaZero has already shown the effectiveness of MCTS in iterative, local policy optimization in discrete action spaces. In our work, we further extend this to continuous action spaces.
>
> == Comparison with other planning embedding approaches ==
>
> While there is some work in incorporating planning into learning (cited in our paper [11]), to the best of our knowledge, this is the first work that aims to use MCTS generated samples in a state-of-the-art on-policy learning (PPO).
>
> == Technical Details ==
>
> We apologize for not being clear on the technical details. To build a tree for continuous domains, we rely on the pre-trained policy for sampling a limited set of **promising** actions (with high probability). The description in Section 3 expound on the details of how we use MCTS in our framework. To be more specific, it clarifies, how we use policy to sample the actions for MCTS, how we select actions at each node of the tree (each node represents a state from the environment) and finally how the immediate reward is incorporated into MCTS search paths. The number of actions per node represents the number of samples we take from policy for a given state.
>
> == Total Computation Time ==
>
> As opposed to stochastic search (one action per state), MCTS searches multiple paths and takes more time. For this reason, we start from a pre-trained policy to limit the number of actions to search. On top of that, we use a batched environment (https://arxiv.org/pdf/1709.02878.pdf) to improve the runtime. We clarify the total runtime for both approaches in the paper.

---

### Official Review · AnonReviewer4 · 2019-11-02
**Official Blind Review #4**

**Rating:** 1

**Review:**

This paper proposes Tree search Policy Optimization (TPO) algorithm for tasks with continuous action spaces. TPO works after a well trained policy can be obtained (PPO is used in the paper). After a well trained policy is obtained, TPO firstly uses the policy to do Monte Carlo Tree Search (MCTS), with at most 32 actions at each node (state) in the tree (to make MCTS work for continuous action spaces). Then after searching, policy is updated by matching the statistics of MCTS policy (mean and variance), since usually MCTS induces a better policy than the current sampling policy. Experiments on MuJoCo tasks show that TPO has performance improvement over PPO.

1. This paper is basically a special case of dual policy iteration methods (as mentioned), with some heuristics to make MCTS work in continuous action spaces. Unlike dual policy iteration, there is no theoretical justification why doing these heuristics are good/convincible.

2. As a paper focusing on experiments, the results are not enough, in the following senses,

a) There are other existing work of MCTS in continuous action spaces, but they are not mentioned, like
"Monte Carlo Tree Search in Continuous Action Spaces with Execution Uncertainty", Yee et al., 2016.
"Monte Carlo Tree Search for Continuous and Stochastic Sequential Decision Making Problems", Couetoux, 2013.
Without comparing with other baselines, the current experiments are not convincing to claim the proposed TPO method is a better choice over other existing methods.

b) The experiments only show application of TPO on PPO, which does not indicate that TPO can improve performances of other methods, like Soft Actor-Critic (SAC), and "Relative Entropy Regularized Policy Iteration", Abdolmaleki et al, 2018 (RERPI), which are known as better choices for Mujoco tasks. Actually, I was wondering how can TPO work with other true off-policy algorithms like the mentioned SAC and RERPI.

c) The learning results/curves seem to be not efficient. The maximum timestep is 4M. What is timestep? Is it the number of iterations? Does that mean the environment steps are totally 4M *32 = 128 M (since every step there are 32 envs in parallel). Please clarify this, and also show how many environment steps (totally how many number of actions have been taken) are there for PPO here. If their environment steps are not the same, then this comparison is not fair (TPO actually uses more actions). And if 128 M environments are used, then the learning is quite inefficient, for example, the final score for Ant of TPO is about 5000, while SAC achieves 6000 after 3 M environment steps.

3. The MCTS policy update stage (second stage) uses mean and variance matching to update policy Eq. (8). I did not see why this objective is good. There is no intuition or comparison to support it. For the Gaussian policy here, KL divergence between MCTS policy and the current policy induces another different matching of mean and variance. How is this objective compared with KL? Please use experiments to justify.

4. TPO works under restricted requirements (require well performed policies), which makes the claimed contribution of "making MCTS work for continuous action spaces" weak (nearly not hold). Actually, the experiments show that if rho = 0.1, there is no good enough policy, then the proposed method does not work at all. This means the contribution is "MCTS works for continuous action spaces in special cases, that a good policy can be used for sampling". The 32 branch factor is enough also because of this reason. The MCTS in discrete action space is guaranteed to converge (UCT algorithm). However, here there is no evidence that this proposed method will achieve similar results if it starts from initialization rather than well trained policies. From this perspective, I consider this method not really a method that "makes MCTS work for continuous action spaces."

5. What is the replay buffer size for MCTS? Are the trajectories in MCTS buffer going to be used again (or thrown away after calculate the mean and variance of MCTS policy)? I suppose the simulations will be just used for once and the MCTS tree will be thrown away (next iteration a new tree will be constructed). If this is the case, then the proposed method is not really a true "off-policy" method as claimed. It does not have the same sample efficiency as other off-policy methods (like SAC, DQN, DDPG), and it does not need to face the same difficulty like importance ratio corrections. Therefore, it is more a policy update step within dual policy iteration framework, rather than true off-policy learning (with replay buffer storing trajectories, and those trajectories will be reused for multiple times.)

6. TPO requires knowledge of maximum timestep, meaning it is not an any-time algorithm (unlike most existing algorithms).

Overall, this paper has no contributions on theory. And I found the experiments cannot show: 1) the proposed TPO is better than other baselines; 2) TPO can work with other methods, especially off-policy algorithms; 3) the learning efficiency, proposed objective/algorithm are questionable; 4) the proposed method is not really a method that makes MCTS work for continuous action spaces, except with restricted requirements.


=====Update=====
I have read the rebuttal and I keep my rating, since there is no revision to see any improvements.

**Experience Assessment:**

I have published one or two papers in this area.

**Review Assessment: Checking Correctness Of Derivations And Theory:**

I carefully checked the derivations and theory.

**Review Assessment: Checking Correctness Of Experiments:**

I carefully checked the experiments.

**Review Assessment: Thoroughness In Paper Reading:**

I read the paper thoroughly.

---

> ### Author Response · Authors · 2019-11-15
> **Response to all Reviewers #4**
>
>
> == Comparison with other MCTS Approaches ==
>
> Thank you for bringing these papers to our attention. We will make sure to cite them in the related work.
>
> “Monte-Carlo Tree Search vs. Model-Predictive Controller: A Lane-Following Example”
> “Monte Carlo Tree Search for Continuous and Stochastic Sequential Decision Making Problems”
>
> These papers proposed to use MCTS solely for planning, whereas our main contribution is the seamless integration of MCTS-guided planning and policy optimization.
>
> In “Deep Reinforcement Learning in Continuous Action Spaces: a Case Study in the Game of Simulated Curling”, the authors do not explore the implications of having environments with immediate reward in MCTS and followed a similar environment setting as AlphaZero. As such, it makes it hard to have a direct comparison with their approach.
>
> In “Continuous Control Monte Carlo Tree Search Informed by Multiple Experts”, as mentioned in the limitations, the authors explored incorporating MCTS search in a supervised setting, as opposed to using MCTS samples in a reinforcement learning setting, particularly for policy iteration methods.
>
> == Sample Efficiency ==
>
> We agree with the reviewer. The total number of samples is higher compared to PPO. However, as we see in Figure 2-a, 2-c, and 2-e, PPO results have already converged and we did not see any further improvement even with more samples. To further reduce the sample complexity, we are working to incorporate a model-based approach in TPO. That is, we iteratively learn a model during the training process and rely on the learned model to search in MCTS.
>
> == Loss Function ==
>
> Our intuition for using this loss function was to minimize the distance between the distribution parameters (policy distribution and estimated distribution from MCTS). We found this loss to be less computationally intensive compared to other approaches. We can replace our loss with a more expensive KL divergence on multidimensional gaussian distributions.
>
> == Replay Buffer ==
>
> At each iteration, we add 2048 samples to the replay buffer. Since we are using PPO, we clear the buffer after multiple iterations of training. However, for other policy iteration algorithm we can reuse the existing data in the replay buffer. We apologize for the confusion regarding the usage of the “off-policy” term.
>
> == MCTS for Continuous Action Spaces ==
>
> We do not claim that TPO works in any environment. Instead, we show that using a pre-trained policy can significantly improve exploration efficiency.  Our contribution is to take a policy in a continuous action space and improve it by generating samples from an expert (MCTS in our case).
>
> == Maximum Timestep ==
>
> We do not rely on knowledge of the maximum timestep in expanding the tree. Instead, we rely on whether a `*done*` signal has been triggered by the environment. This is similar to other policy iteration algorithms like PPO or SAC.
>
> == Comparison with other Policy Iteration Methods ==
>
> In this work, we show that MCTS can indeed help to improve the quality of results significantly, even in continuous action spaces using a SOTA on-policy algorithm. As you suggested, we are working to add more policy iteration algorithms (e.g. SAC and RERPI) into our framework.

---

### Author Response · Authors · 2019-11-15
**Response to all Reviewers**

We would like to thank the reviewers for their insightful comments. We are glad that they feel we have “promising results” and have tackled an “interesting research problem”. Their valuable comments have motivated us to continue in this research direction and to further improve the quality of the paper.

---

### Decision · Program_Chairs · 2019-12-19

**Decision:**

Reject

**Comment:**

The paper proposes a tree search based policy optimization methods for continuous action state spaces. The paper does not have a theoretical guarantee, but has empirical results.

Reviewers brought up issues such as lack of using other policy optimizations methods (SAC, RERPI, etc.), sample inefficiency, and unclear difference with some other similar papers. Even though the authors have provided a rebuttal to address these issues, all the reviewers remain negative. So I can only recommend rejection at this stage.